# NerfBaselines: Consistent and Reproducible Evaluation of Novel View Synthesis Methods

**Jonas Kulhanek**[1,2]**, Torsten Sattler**[1]

[1]Czech Technical University in Prague, Czech Institute of Informatics, Robotics and Cybernetics
[2]Czech Technical University in Prague, Faculty of Electrical Engineering

## Abstract

Novel view synthesis is an important problem with many applications, including AR/VR, gaming, and robotic simulations. With the recent rapid development of Neural Radiance Fields (NeRFs) and 3D Gaussian Splatting (3DGS) methods, it is becoming difficult to keep track of the current state of the art (SoTA) due to methods using different evaluation protocols, codebases being difficult to install and use, and methods not generalizing well to novel 3D scenes. In our experiments, we show that even tiny differences in the evaluation protocols of various methods can artificially boost the performance of these methods. This raises questions about the validity of quantitative comparisons performed in the literature. To address these questions, we propose NerfBaselines, an evaluation framework which provides consistent benchmarking tools, ensures reproducibility, and simplifies the installation and use of various methods. We validate our implementation experimentally by reproducing the numbers reported in the original papers. For improved accessibility, we release a web platform that compares commonly used methods on standard benchmarks. We strongly believe NerfBaselines is a valuable contribution to the community as it ensures that quantitative results are comparable and thus truly measure progress in the field of novel view synthesis. **Web:** https://nerfbaselines.github.io

## 1 Introduction

Benchmarks such as ImageNet [12], KITTI [19], COCO [35], Middlebury [55] *etc.* are a major driving force behind computer vision research. In particular, using community-wide accepted evaluation protocols has accelerated research as fair comparisons have been made much easier (no need to reimplement, tune parameters, *etc.*). It has also made it easier to measure and follow progress on tasks (such as object detection or localization). Measuring progress by comparing numbers is only possible if everyone follows exactly the same evaluation protocol down to the last detail. Without this great attention to detail, numbers in tables might only create the illusion of a fair comparison, as small changes in the evaluation protocol can severely impact the ranking of methods. This is illustrated in Figure 1 in the area of novel view synthesis, where a seemingly innocuous change in the evaluation protocol (manually downscaling larger images instead of using the provided downscaled JPEGs) can significantly boost the numbers of a middling approach. In other words, without a standardized evaluation protocol, numbers in tables potentially only create the illusion of progress, as better numbers might also be a product of changes in the evaluation protocol made by a set of authors (as published work typically does not describe the evaluation setup in too much detail). At the same time, it opens the door for malicious actors to tune the protocol rather than making actual progress, as the former might be easier than the latter.

The field of photorealistic 3D reconstruction has seen explosive growth recently, first based on NeRFs [44], later on 3DGS [23], with multiple papers published on arXiv per day. Unfortunately, there hasn't been time to establish common evaluation protocols that are followed by everyone.

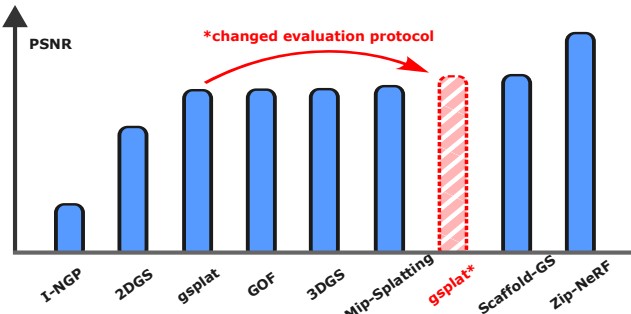

Figure 1: **Impact of altering evaluation protocol.** By changing how images were downscaled, Gsplat [71] increased its rank by 3 places in PSNR on the Mip-NeRF 360 dataset [3].

The differences in evaluation protocols may include: using different downscaling strategies (as demonstrated in Figure 1), or different image resolutions, whether images are undistorted, which type of LPIPS [79] and which version was used, which background color was used when blending semi-transparent images, whether images are rounded to the `uint8` range, image range normalization prior to metrics computation, *etc.*

There are efforts to create unified development environments for NeRFs and 3DGS, *etc.* (*eg.* Nerf-Studio [60], SDFStudio [74]). These development environments are designed to make it easier to develop new methods by providing common building blocks such as layers, losses, data loaders, *etc.* This typically includes evaluation protocols. While these frameworks offer re-implementations of popular methods, these re-implementations do not always faithfully reproduce the results of the original implementations (*cf.* Tab. 1 for the case of NerfStudio). Another issue is that methods have to be implemented within the development environment, which is time-consuming and makes wide-scale adoption difficult. In addition, as development frameworks change over time, third-party contributions of baselines might become unusable if they are not constantly adapted to the changes in the framework, including technical details such as updating libraries on which the software depends. As a result, such development frameworks are not a real solution to the problem of reliable and consistent benchmarking of methods for novel view synthesis.

This paper aims to provide a framework for consistent and reproducible evaluation of novel synthesis methods. To this end, we present a software framework that makes it easy to: **1)** add new methods by providing a simple API for adding them while also providing encapsulation that allows us to easily install the original software with all its (by now potentially outdated) libraries by careful curation. Our framework, NerfBaselines, already implements many of the most popular approaches for various different subproblems (in-the-wild [40], underwater [33], *etc.*). **2)** add new datasets and evaluate existing methods on additional datasets by providing a unified dataset representation. At the same time, the framework integrates many of the most popular datasets (Blender [44], LLFF [43], Mip-NeRF 360 [3], Photo Tourism [58], Tanks and Temples [28], SeaThru-NeRF [33], NeRF On-the-go [54], NerfStudio [60]). **3)** provides a single evaluation protocol per supported dataset, enabling fair comparisons between different methods. **4)** in addition, we contribute a website that collects results under the evaluation protocol, including visualization and rendering capabilities.

The development of NerfBaselines was a significant engineering effort – designing a software framework that is compatible with the most popular development frameworks, and integrating existing approaches into our framework (a significant effort since in most cases, the original installation instructions were not valid anymore and it required identifying library versions still compatible with the software), developing visualization and other tools, *etc.*. As with any good evaluation framework, the exact implementation and design choices will be of little interest to most researchers.[1] As such, this paper only briefly summarizes the NerfBaselines framework. Rather, the focus of this paper is on demonstrating the usefulness and value of NerfBaselines. We believe that the insights presented in the paper are of interest and value to the community. Our main insights are:

---

[1] Arguably, the purpose of evaluation frameworks and benchmarks is to remove the need for the researchers to become intimately familiar with design choices and implementation details. Part of the purpose of evaluation frameworks such as NerfBaselines is that the effort spent in developing them removes the need for other researchers to invest time to get baselines to run and to investigate source codes to ensure fair comparisons.

1. We show that the published results of most of the popular methods can be (closely) reproduced by NerfBaselines. This shows that malicious authors trying to boost the performance of their methods by gaming the evaluation protocol are so far not a major issue and that the numbers reported in published papers can be (largely) trusted.

2. We show that there are exceptions where certain design choices significantly impact the reported performance. While the method ranking is largely unchanged when switching from one protocol to another, the differences in performance of the same method under different protocols can exceed the difference between methods under the same protocol. This shows that one can create the illusion of better performance by using a slightly different evaluation protocol than others. This result clearly demonstrates the need for a consistent and fair evaluation that NerfBaselines offers to the community.

Furthermore, we show that it is easy to evaluate existing methods on more datasets by integrating these datasets into NerfBaselines. As a service to the community, we provide more complete results on some datasets that are used by some but not all popular methods. We also demonstrate the value of visualizing tools (the viewer and the camera trajectory editor included in NerfBaselines) by showing how various methods behave differently as the camera moves further from the training trajectory.

## 2 Related work

In recent years, progress in novel view synthesis has enabled real-time photo-realistic rendering of images from novel camera viewpoints. There has been a surge of interest, first with the advent of neural radiance field methods (NeRFs) [44, 36, 2, 3, 78, 18, 52, 16, 59, 69, 29, 8, 17, 53, 46, 4, 64, 48, 51, 40, 5, 49, 25, 27, 62, 21, 63, 65, 70, 11, 50, 73, 7] which enabled photo-realistic results and a wide range of applications. The second wave of attention came with the introduction of 3D Gaussian Splatting (3DGS) [23, 15, 81, 75, 37, 24, 47, 22, 72, 76, 14, 41, 67, 38, 80, 9], matching NeRFs in terms of the rendering quality while enabling real-time rendering.

### 2.1 Existing codebases

Most current methods are based on a few core repositories: NerfStudio [60], Multi-NeRF [2, 3], Instant-NGP [46] for NeRFs, and Gaussian Splatting [23] for 3DGS, typically with moderate modifications. Therefore, in NerfBaselines, we focused on integrating these core repositories, as it simplifies the subsequent integration of derivative works. In Figure 2, we illustrate the relationships between popular repositories and highlight those currently integrated with NerfBaselines.

NerfStudio [60]
  └─ Tetra-NeRF [30], Instruct-NeRF2NeRF [21], LERF [25], Splatfacto [71]
Multi-NeRF [3, 2]
  ├─ Zip-NeRF, Mip-NeRF [2], RawNeRF [45]
  └─ MipNeRF-360 [3]
        └─ SeaThru-NeRF [33], NeRF on-the-go [54]
Gaussian Splatting (INRIA) [23]
  ├─ Mip-Splatting [75]
  │     └─ Gaussian Opacity Fields [76]
  ├─ WildGaussians [31], GS-W [77], ScaffoldGS [37], 3DGS-MCMC [26]
  └─ 2D Gaussian Splatting [22], AbsGS [72], Taming-3DGS [39]
NeRF-W (PL reimplementation) [44], NeRF [44], Instant-NGP [46]
K-Planes [17], TensoRF [8], gsplat [71], Plenoxels [16]

Figure 2: **Existing codebases.** Integrated methods are **bold green**.

**NerfStudio** is a popular framework that introduced the modular separation of NeRFs into components such as ray samplers, radiance field heads, etc. It supports various dataloaders, camera types, and export formats.

**Multi-NeRF** is fully implemented in JAX and does not have custom CUDA kernels (unlike Instant-NGP or NerfStudio), making it easy to install, but slower. Early versions based on Mip-NeRF360 supported only a single camera per dataset and a single image size; therefore, in NerfBaselines, we have extended these methods to handle more complex datasets.

**Instant-NGP** is a highly optimized implementation that has inspired numerous follow-up works [4, 60, 25, 27]. However, it is a less popular choice as a codebase due to its C++ training code being more difficult to extend.

| Method (Scene) | Paper | Public | NerfStudio | NerfBaselines |
|---|---|---|---|---|
| TensoRF (lego) | 36.46 | 36.54 | 33.09 | 36.49 |
| 3DGS (garden) | 27.41 | 27.39 | 27.17 | 27.34 |
| Instant-NGP (lego) | 36.39* | 36.09* | 15.24 | 35.65 |
| SeaThru-NeRF (Panama) | 27.89 | 27.85 | 31.28 | 27.82 |
| Zip-NeRF (garden) | 28.20 | 28.22 | – | 28.19 |

Table 1: **Implementations reproducibility.** We compare the PSNR reported in the paper with the PSNR of the official implementation, the PSNR of NerfStudio's implementation, and NerfBaselines's integration of the same method. *Instant-NGP uses black background (NerfBaselines white).*

**Gaussian Splatting (INRIA)** is the most popular choice for 3DGS methods, primarily because (until recently) it was unmatched in performance. The repository only supports pinhole cameras with their centre of projection being in the centre of the image. We have extended it to work with arbitrary camera models, performing undistortion/distortion for more complicated camera models.

## 2.2 Benchmarking frameworks

There exist some works proposing a unified platform for various methods to be implemented [60, 74, 20]. Out of these, NerfStudio [60] is perhaps the closest to our framework in that it offers unified data loading and evaluation tools for multiple novel view synthesis methods. It is meant to provide building blocks when developing new NeRF methods (it implements various ray samplers, radiance field heads, etc.). However, each method needs to be implemented from scratch to be able to use NerfStudio. Currently, the NerfStudio-reimplemented methods have performance different from the original implementation as can be seen in Table 1. In contrast, in NerfBaselines, the integrated methods still use the original released source code, and only a small interface is written to interface with the codebase. Another difference is that while NerfStudio is more focussed on practical applications and fast development, NerfBaselines focuses on fair, reproducible, and consistent evaluation. Therefore, the design choices differ. Finally, we also provide a web benchmark platform where new methods are constantly being added and can be compared.

## 3 NerfBaselines framework

When designing the evaluation framework, we followed these constraints: **1)** *Easy integration of new methods*, to ensure the platform can grow and remain useful. **2)** *Matching official results*, so reported numbers are consistent. **3)** *Support for new data/scenarios*. Many official implementations only work with specific datasets or assumptions (*eg.*, identical intrinsics). NerfBaselines removes such limitations. **4)** *Stability and reproducibility over time*. Methods should remain installable and yield consistent results long after their release. To simplify integration, NerfBaselines uses official implementations without reimplementation. Each method wraps the original code using a common **API**, making it easier to preserve official behavior and apply methods to new datasets. The API separates model logic from data loading and evaluation, enabling reuse and consistent evaluation. For long-term stability, we freeze code and dependencies for each method and use isolated environments (Docker, Apptainer, or Conda) to ensure reproducible installations. Retraining every method is inefficient and resource-heavy. Instead, we provide an **online benchmark**[2] with reported results, downloadable checkpoints and predictions, and 3D reconstructions. See the video in *supp. mat.*

**Unified API.** Existing codebases [60, 3, 23, 8, 23] typically consist of: **model implementation**, which optimizes the scene and renders views; **dataloader**, for parsing datasets; and **evaluation code**, for computing metrics. Unfortunately, these parts are often tightly coupled and vary in structure, making reuse difficult and comparison inconsistent.

In NerfBaselines, we isolate the **model implementation** and use standardized components for loading and evaluation. This allows us to apply any method to any dataset (with a shared structure) and ensures all methods follow the same evaluation protocol. We identified a shared structure across both raycasting-based methods [60, 3, 23, 8, 68] and rasterization-based methods [23, 24, 75, 76, 71].

---

[2] https://nerfbaselines.github.io/

Each method defines a `method` class with functions like `train_iteration` for performing one training step and `render` for rendering single frame. Methods with appearance conditioning [40, 60] also implement appearance embedding extraction. The interface allows methods to be integrated via a thin wrapper calling the original code, ensuring correctness and minimal effort The interface (details in the *supp. mat.*) was chosen to make it easy to integrate the methods as one needs to write only a thin wrapper that calls the official code rather than reimplementing the method within NerfBaselines. This also makes it easy to ensure the integrated method matches the official code as the official code is called by the wrapper.

**Environment isolation.** Another requirement of an evaluation framework is long-term stability. Methods often have many dependencies that are poorly specified, making installation impossible over time due to version conflicts or missing packages. When designing NerfBaselines, we found that 12 out of 19 methods could not be installed using the official instructions, often due to unspecified or outdated dependencies. Additionally, codebases change: 1) released checkpoints may become incompatible with newer code, or 2) reported numbers may no longer match. To ensure stability and reproducibility, we freeze both the source code and dependencies for each method. Each method is installed in its own isolated environment, avoiding conflicts and keeping the user's setup clean.

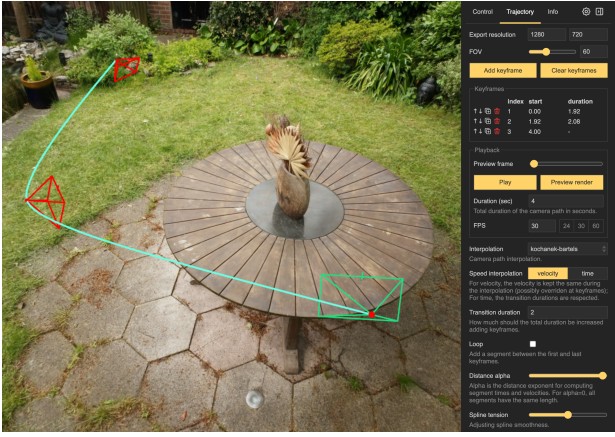

Figure 3: The **NerfBaselines Viewer** enables interactive rendering, shows train/test cameras, and input point cloud. The figure shows trajectory editor used to for rendering custom camera trajectories.

Communication with these environments is done via interprocess communication. NerfBaselines handles dependency installation the first time a method is run. We support three levels of isolation: Conda [1], Docker [42], and Apptainer [32]. Users can choose the backend that best fits their system and isolation needs (e.g., HPC setups)

**Viewer & camera trajectory rendering.** In novel view synthesis, the evaluation is often conducted on a set of test images, typically a subsequence of the training views. Unfortunately, this approach does not adequately demonstrate the method's robustness and multiview consistency in capturing the 3D scene [34]. Rendering a video from a custom trajectory from truly novel viewpoints farther away from the training images often provides an insightful evaluation. While the unified interface presented in Section 3 'Unified API' significantly simplifies the process of rendering such trajectories, we further enhance this capability by providing an interactive viewer. This viewer allows users to inspect method's performance outside the training camera distribution. It also includes a camera trajectory editor for designing custom trajectories and video rendering. The editor is shown in Figure 3.

**Web platform.** Finally, it would be costly and environmentally unfriendly to have to retrain all methods whenever researchers want to compare with other methods. Therefore, we release an **online benchmark**, where the methods can be compared on different datasets. The web platform allows researchers to download the rendered images or the checkpoints that reproduce the numbers. For selected methods, the web platform can visualize the reconstructed 3D scene online. We show the web platform in the *supp. mat.*

## 4 Evaluation protocols

To ensure a fair evaluation of various methods, it is important to use the same evaluation protocol. Otherwise, as we show in the experimental section, even small deviations from the evaluation protocol can lead to changes in the reported numbers. This often renders the comparisons unfair. As stated in the previous section, the main objective of NerfBaselines is to standardize the evaluation protocols such that researchers can safely compare with other methods without **a)** risk of an unfair evaluation, and **b)** without having to spend a lot of time and effort on matching the details of the evaluation

protocols. While the details of the evaluation protocols are standardized in the NerfBaselines code, here we describe them in detail and motivate the design choices.

**1) Images are evaluated in the `uint8` range.** The reason for this is reproducibility: every published method should store and publish its predictions on the test set. This allows the results to be verified and enables future computation of new metrics as well as qualitative comparisons, *etc.* These predictions are stored in `uint8`, which is sufficient for most datasets (considering that `uint8` images were used for training), while `fp32` images would require significantly more memory. Furthermore, almost all publications have released either no predictions or only `uint8` image predictions, and evaluation protocols should therefore be compatible with this design choice in order to use this data and allow comparison with existing methods. Consequently, NerfBaselines rounds the predictions to the `uint8` range during evaluation. In our experiments, this can cause a small but noticeable difference (approximately $\pm 0.02$ PSNR).

**2) LPIPS details.** A common issue in literature is not to specify the details of the LPIPS metric. Older papers mainly use VGG, while some recent ones opt for AlexNet-based LPIPS. The benefit is that AlexNet is more lightweight and faster to evaluate. Therefore, we use AlexNet by default, but for older datasets we use VGG. In NerfBaselines, we use the torch-based implementation with the checkpoint version $0.1$.

**3) Using pre-downscaled images.** We argue for the use of pre-downscaled released images, because the default downscaling algorithms differ for different platforms and libraries. Loading larger images and downscaling them before evaluation could yield different results depending on platform - which is exactly what the unified evaluation protocol should avoid.

**4) PSNR & SSIM details.** Again, many choices are possible and we just need to make sure the exact same code and parameters are used. This is not the case with some existing libraries (e.g., dm-pix [10], scikit-image [61], torchmetrics [13]), where the SSIM defaults can differ. Our choice (default for skimage) is predominantly used in the literature, hence we follow the same practice. In order to compute the metrics, we convert the images to `float32`, $[0, 1]$ range. PSNR is then computed using the following formula: $-10 \cdot \log_{10}\left(\sum_i (x_i - y_i)^2\right)$. Note, in some implementations [13], the data is first normalized to from $[\min_i x_i, \max_i x_i]$ to $[0, 1]$. This is not done in NerfBaselines. To compute SSIM [6], we use the following parameters: kernel size $= 11$, $\sigma = 1.5$, $k_1 = 0.01$, $k_2 = 0.03$. Again, we assume that the data are in the $[0, 1]$ range and do not perform normalization. The SSIM is averaged over the image and over channels (equivalent to `multichannel` in scikit-image [61]).

**Dataset specific details**. For the Blender [44] dataset, we use VGG-based LPIPS as it was used in the original publication [44]. We blend the transparent images with white background. In Mip-NeRF [3], we use the original $4\times$ downscaled `JPG` images for outdoor and the original $2\times$ downscaled `JPG` images for indoor scenes. We also use VGG-based LPIPS. For the LLFF dataset [43], we also use VGG-based LPIPS. For the Photo Tourism dataset [58], we use the NeRF-W [40] evaluation protocol, where (after the training is finished) image appearance embeddings are optimized on the left half (rounded up) of the images and the metrics are computed on the right half (rounded down).

# 5 Experiments

In our experiments, we **1)** motivate the need for standardized evaluation protocols by showing how small differences can lead to inconsistent results and erroneous conclusions, we then **2)** verify that the integrated methods match the original metrics, **3)** demonstrate transferability to novel datasets, and **4)** show an application of NerfBaselines to qualitatively compare methods outside training trajectories. All our experiments used NVIDIA A100 GPUs. A single GPU was used for all but Mip-NeRF 360 [3], which used four GPUs.

## 5.1 Importance of unified evaluation protocol

We motivate the need for our benchmarking framework by showing how small differences in evaluating protocols can lead to differences in the presented results. We specifically pick differences in the evaluation protocols used by some methods. We show the difference on three datasets: Mip-NeRF 360 [3], Blender [44], and Photo Tourism [58]. For all datasets, we report the performance under the official NerfBaselines evaluation protocol ($P_1$), an alternative evaluation protocol ($P_2$), and the

| | $P_1$ PSNR | $P_1$ rank | $P_2$ PSNR | $P_2$ rank | $P_2$ one-in | $P_1$ one-in |
|---|---|---|---|---|---|---|
| Zip-NeRF [4] | 28.55±0.01 | 1 | 28.84±0.01 | 1 | 1 | 1 |
| Scaffold-GS [37] | 27.71±0.01 | 2 | 28.01±0.02 | 2 | 2 | **4** |
| Mip-Splatting [4] | 27.49±0.03 | 3 | 27.78±0.04 | 3 | **2** | **6** |
| Gaussian Splatting [23] | 27.43±0.02 | 4 | 27.68±0.03 | **5** | **3** | **6** |
| GOF [76] | 27.42±0.03 | 5 | 27.71±0.02 | **4** | **2** | **6** |
| Gsplat [71] | 27.41±0.02 | 6 | 27.68±0.03 | **5** | **3** | 6 |
| 2DGS [22] | 26.81±0.03 | 7 | 27.11±0.03 | 7 | 7 | 7 |
| NerfStudio [60] | 26.39±0.03 | 8 | 26.57±0.01 | 8 | 8 | 8 |
| Instant-NGP [46] | 25.51±0.04 | 9 | 25.75±0.03 | 9 | 9 | 9 |

Table 2: **Mip-NeRF 360 evaluation protocol differences.** We report PSNR ($\pm$ standard deviation) under protocols $P_1$ (official), and $P_2$ (manual downscaling). Changes in ranking denoted in **bold**.

| | $P_1$ PSNR | $P_1$ rank | $P_2$ PSNR | $P_2$ rank | $P_2$ one-in | $P_1$ one-in |
|---|---|---|---|---|---|---|
| GOF [76] | 33.45 | 1 | 33.76 | **2** | 1 | **4** |
| Mip Splatting [75] | 33.33 | 2 | 33.85 | **1** | 1 | **4** |
| Gaussian Splatting [23] | 33.31 | 3 | 33.76 | **2** | 1 | **4** |
| Scaffold-GS [37] | 33.08 | 4 | 33.48 | 4 | **1** | 4 |
| Instant-NGP [46] | 32.20 | 5 | 32.70 | 5 | 5 | 5 |

Table 3: **Blender evaluation protocols comparison.** We report PSNR under protocols $P_1$ (official), and $P_2$ (black background). Bold numbers denote changes in ranking.

ordering under both protocols. We further show $P_2$ ($P_1$) 'one-in' results, where we show the rank of the method as if it was the only one using $P_2$ ($P_1$), while all other methods used $P_1$ ($P_2$).

**Mip-NeRF 360 dataset.** The Mip-NeRF 360 dataset comes with the original large-scale images and the images which were downscaled by the authors and saved as JPGs. While original NeRF methods used the downscaled images, in the official source code of 3D Gaussian Splatting [23], the default loader uses the large-scale images that are downscaled on the fly. The authors report both numbers for manually scaled images and when using the released downscaled images, however, some follow-up works [75, 76, 22] already use the manual downscaling only when comparing with prior work. In Table 2, we show the PSNR averaged over all scenes of the Mip-NeRF 360 dataset under protocols $P_1$ (official NerfBaselines one), and $P_2$ (when manually downscaling the large images). We also report the standard deviation computed over four independent trainings of each method. As can be seen from the results, while the ranking is consistent for NeRF-based methods, for 3DGS-based methods the ranking changes when changing the evaluation protocol. Furthermore, suppose one of the 3DGS-based methods chooses a different evaluation protocol. In that case, it can become better than other 3DGS-based methods and similarly, all 3DGS-based methods except for Scaffold-GS [37] can become the worst by following the 'official' evaluation protocol if all other methods use $P_2$.

**Blender dataset.** The Blender dataset [44] contains RGBA images with a transparent background. In the original NeRF paper, authors used white color as the background ($P_1$), however, in Instant-NGP, a black background color was used ($P_2$). In Table 3, we show the PSNR averaged over all Blender scenes. The results show, that if any 3DGS-based method (by accident) uses $P_2$, it would seem like it outperforms all 3DGS-based methods. Even the relative ordering under $P_1$ changes under $P_2$.

**Photo Tourism dataset.** Since the dataset contains images with varying appearances, it is important to adapt to the appearance of the test images at test time. Therefore, in the NeRF-W [40] paper, the evaluation protocol was standardized as using the left half of each test image to optimize the image's appearance embedding, and to compute the metrics on the (previously unseen) right part. However, some methods use a different evaluation protocol where they optimize the representation on the full test images [77, 66]. In Table 4, we show the PSNR averaged over scenes Trevi Fountain, Brandenburg Gate, and Sacre Coeur of the Photo Tourism dataset [58]. Protocol $P_1$ is the official NerfBaselines one (NeRF-W [40]), and $P_2$ uses full test images when optimizing appearance embeddings. As expected, this protocol change has a large impact when only one method uses $P_2$. Interestingly, the order changes when all methods use $P_2$.

| | $P_1$ PSNR | $P_1$ rank | $P_2$ PSNR | $P_2$ rank | $P_2$ one-in | $P_1$ one-in |
|---|---|---|---|---|---|---|
| WildGaussians [31] | 24.56 | 1 | 25.96 | **3** | 1 | **4** |
| Gsplat [71] | 23.66 | 2 | 26.28 | **1** | 1 | **5** |
| Scaffold-GS [37] | 23.50 | 3 | 25.97 | **2** | 1 | **6** |
| NeRF-W *re.* [40] | 21.75 | 4 | 25.61 | 4 | 1 | **6** |
| GS-W [77] | 21.38 | 5 | 23.55 | **6** | 3 | **6** |
| K-Planes [17] | 21.10 | 6 | 23.98 | **5** | 2 | 6 |

Table 4: **Photo Tourism evaluation protocols comparison.** We report PSNR under protocols $P_1$ (NeRF-W), and $P_2$ (full test images). Bold numbers denote changes in ranking.

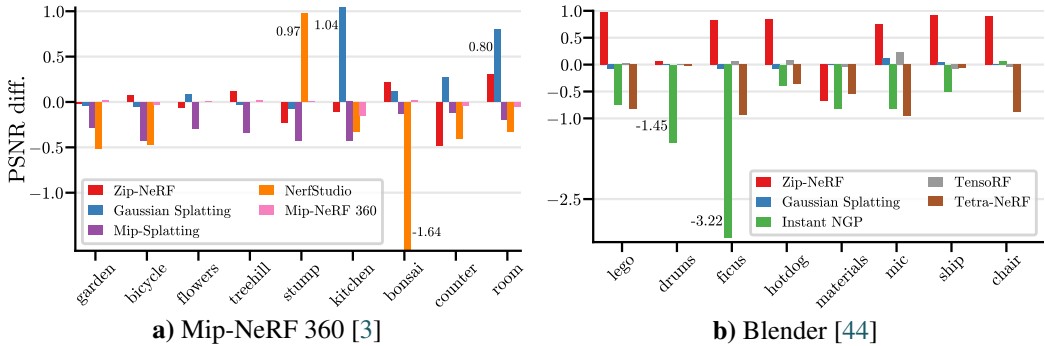

**a)** Mip-NeRF 360 [3]    **b)** Blender [44]

Figure 4: **Mip-NeRF 360 [3] and Blender [44] results** comparing PSNRs obtained via NerfBaselines with those reported in the original papers. We show the difference in PSNR. In most cases, the difference is $< 1\%$. Instant-NGP [46] and Mip Splatting [75] are consistently underperforming because different evaluation protocols were used in the papers.

## 5.2 Reproducing published results

To verify that our framework can reproduce the results from the papers, we reevaluate important methods on the standard benchmark datasets: Mip-NeRF 360 [3] and Blender [44]. We use the same evaluation protocol for all methods. The results are compared to the original numbers as published in the papers in Figure 4, with detailed numbers given in the *supp. mat.* Note, that we only compare with methods that released their numbers on the datasets in the corresponding publications.

**Mip-NeRF 360 results.** As shown in Figure 4, NerfBaselines reproduces the original results with a deviation of less than 1% for most scenes. For Mip-Splatting [75] and 3DGS [23], the difference in the numbers was caused by the different evaluation protocols used as discussed in Section 5.1. In the case of NerfStudio [60] and Tetra-NeRF [30], the codebase evolved since the time of the release which is likely the cause of the difference.

**Blender results.** From the results, the discrepancy is again small for most methods. However, for the Instant-NGP method [46], we can notice larger differences in PSNR, especially for 'drums', and 'ficus'. Note, that Instant-NGP [46] uses a black background for training and evaluation which was

| PSNR↑ | barn | caterpillar | truck | lighthouse | playground | train | auditorium | ballroom | courtroom | museum | palace | temple |
|---|---|---|---|---|---|---|---|---|---|---|---|---|
| | | Training Data | | | Intermediate | | | | Advanced | | | |
| Instant NGP [46] | 25.90 | 21.72 | 22.85 | 21.65 | 23.33 | 20.01 | 20.67 | 21.62 | 19.44 | 15.19 | 19.09 | 17.84 |
| NerfStudio [60] | 26.40 | 21.71 | 23.37 | 20.85 | 24.69 | 20.43 | 20.77 | 22.68 | 20.24 | 17.84 | 17.68 | 17.06 |
| Zip-NeRF [4] | 29.26 | 23.94 | 25.09 | 23.07 | 27.13 | 22.19 | 24.52 | 25.45 | 22.17 | 19.34 | 19.11 | 20.58 |
| Gaussian Splatting [23] | 27.51 | 23.38 | 24.25 | 22.11 | 25.37 | 21.67 | 24.13 | 24.07 | 23.12 | 20.92 | 19.63 | 20.85 |
| Mip-Splatting [75] | 27.75 | 23.42 | 24.36 | 22.25 | 25.87 | 21.82 | 24.41 | 24.15 | 23.00 | 20.88 | 19.63 | 20.55 |
| Gaussian Opacity Fields [76] | 25.72 | 21.78 | 22.33 | 21.80 | 23.89 | 19.69 | 23.20 | 22.84 | 21.15 | 19.92 | 16.46 | 20.29 |

Table 5: **Tanks & Temples [28] results.** We show the PSNR of various implemented methods. The first , second , and third values are highlighted.

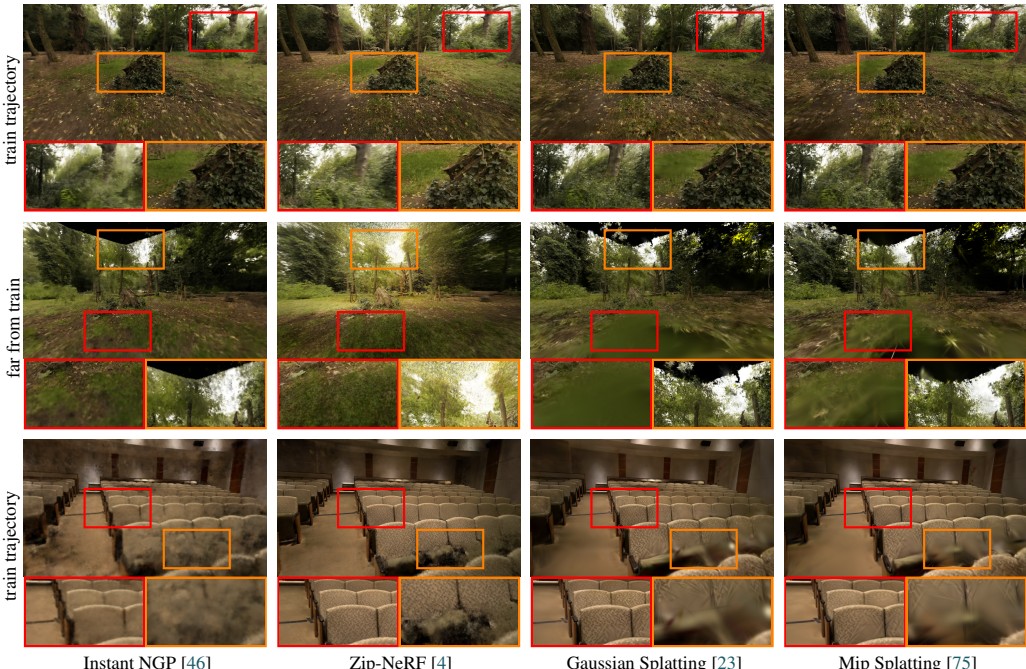

Figure 5: **Qualitative results.** We compare methods on views close and far from the training trajectory. **Top:** MipNeRF360/stump scene, **bottom:** T&T/Auditorium.

shown in the previous section to cause the difference. Since Tetra-NeRF [30] uses NerfStudio [60] and the codebase evolved since the time of the release, we can notice a slight drop in the performance.

### 5.3  Tanks & Temples evaluation

To demonstrate how NerfBaseline simplifies the transfer of existing methods to new datasets, we evaluate various integrated methods on the Tanks and Temples [28] dataset. For the dataset, we run COLMAP reconstruction [56, 57] with a simple radial camera model shared for all images. Afterward, we undistorted and downscaled the images by a factor of 2. For NerfStudio [60], we run the Mip-NeRF 360 configuration (which from our experiments performs better on the dataset than the default configuration). The results are given in Table 5. As we can see, for easier scenes, the reconstructions are dominated by Zip-NeRF [4]. For 'Advanced', there is no single best-performing method. We believe this is caused by NeRFs, with its fixed capacity, not scaling as well to larger scenes as 3DGS, where the capacity is adaptively increased.

### 5.4  Off-trajectory qualitative comparison

While test-set metrics enable an effective way of comparing different methods, they are insufficient to fully evaluate the perceived quality [34]. Rendering images from poses with varying distances from the training camera's trajectory provides a lot of insight into the robustness of the learned representations. Therefore, NerfBaselines provides a viewer and a renderer to enable visualising methods and rendering images/videos outside the train trajectory. In Figure 5, we compare various methods by rendering trained scenes both close to the training camera trajectory and far from it. Notice how in the second row Instant-NGP [46], and 3DGS methods [23, 75] cannot fill the sparsely observed sky, while NerfStudio [60] and Zip-NeRF [4] can achieve it thanks to space contraction. Also, notice how 3DGS methods [23, 75] are more blurred in less observed regions.

## 6  Conclusion

In this paper, we demonstrated how tiny differences in evaluation protocols used in novel view synthesis methods, e.g. NeRFs and 3DGS, can lead to large differences in results, possibly changing the ranking of the methods on public benchmarks. We show that most published numbers of popular

methods can be trusted, but there are exceptions (using different downscaling, background color, *etc.*) which cause sizeable difference in performance. To address the problem, we standardized the evaluation protocols and proposed an evaluation framework, called NerfBaselines, that ensures a fair and consistent evaluation. We validated our framework by showing that the integrated methods reproduce numbers from papers, and showed how it can be used to easily evaluate on new datasets. We strongly believe NerfBaselines is a valuable contribution to the community as it ensures that quantitative results are comparable and thus truly measure progress in the field of novel view synthesis.

**Limitations.** A major concern for each evaluation framework (including ours) is if it will be adopted by the research community. To this end, we made it easy to integrate new methods and datasets by designing the API and using isolated environments, created extensive documentation, and pledge to continuously support and extend the platform.

**Acknowledgments.** This work was supported by the Czech Science Foundation (GAČR) EXPRO (grant no.23-07973X), the Grant Agency of the Czech Technical University in Prague (grant no. SGS24/095/OHK3/2T/13), and by the Ministry of Education, Youth and Sports of the Czech Republic through the e-INFRA CZ (ID:90254). We want to thank Brent Yi for helpful discussions regarding the NerfStudio codebase.

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

# Supplementary Material

In this Supplementary Material, we extend Section 3 from the main paper by providing the full Method API in Section A.1. We attach a video showing the web platform, the interactive viewer, and showing qualitative results on the Tanks and Temples dataset [28]. We describe the attached video in Section A.2. In Section 5.1 in the main paper, we showed how changing the evaluation protocols impacts performance in terms of PSNR. Sec. A.3 expands the results to also show SSIM, and LPIPS. We extend the reproducibility comparison from Section 5 of the main paper in Section A.4 by giving the exact numbers for the already integrated methods. As an addition to the video, we also show screenshots from the web platform in Section A.5. Finally, we provide a brief introduction on how to use NerfBaselines and reproduce the results in Section A.6.

## A.1   Method API

Every method implements the following interface:

- `constructor(train_dataset?, checkpoint?, config_overrides?)`: The constructor takes as its inputs the (optional) training dataset instance (a set of images and camera parameters) or the (optional) checkpoint. At least one of the two has to be provided. Also, it optionally takes as an argument a subset of hyperparameters overriding the method's default hyperparameters.
- `train_iteration`: Performs one training step (using the train dataset), updating the parameters.
- `save(path)`: Saves the current checkpoint.
- `render(camera, options?)`: Renders the 3D scene using the camera parameters with an optional rendering configuration (e.g., including camera appearance embedding).
- `get_info` and `get_method_info`: Returns information about the trained model and the base method, respectively.
- (optional) `optimize_embedding(dataset, embedding?)`: Optimizes the appearance embedding on the single-image dataset (if the method supports it), optionally using `embedding` argument as the starting point.
- (optional) `export_mesh(path, options?)`: Exports the reconstruction as a mesh.

## A.2   Video

The video included in this Supplementary Material [3] is split into several parts:

**Online benchmark.** A core part of NerfBaselines is a web page that evaluates and compares the methods integrated inot NerfBaselines on various datasets. This part of the video shows the functionality of the website: For each method, the numbers on all scenes as well as the averaged numbers are shown. We further show the numbers reported in the original paper and explain discrepancies in evaluation protocols. For every method on every dataset/scene a checkpoint can be downloaded (reproducing the numbers) as well as the set of predictions on the test set. Finally, for some methods, we implement an online viewer where the reconstruction can be visualized – running in the web browser.

**Interactive viewer.** Our interactive viewer is based on viser [60] and enables visualizing trained model. It has a trajectory editor which can be used to create a camera trajectory and render a video.

**Mip-NeRF 360 and Tanks and Temples results.** Finally, we qualitatively compare Gaussian Splatting [23], Mip-Splatting [75], Zip-NeRF [4], Instant-NGP [46], and NerfStudio [60] on custom camera trajectories. We generate trajectories such that we start from regions close to the training trajectory, next we move further away from the scene center, and finally, move close to the scene geometry. The purpose is to visualize how different methods handle these viewpoint changes. We show the results on two scenes from the Mip-NeRF 360 dataset [3] (stump and kitchen), and two scenes from the Tanks and Temples dataset [28] (temple and lighthouse). Notice, how 3DGS behaves well close to the training trajectory but has blank spots outside (where the geometry is missing). On the other hand, NeRFs (especially NerfStudio [60]) is better able to extrapolate to less visible regions further from the scene center.

---

[3] https://nerfbaselines.github.io/video.html

## A.3 Differences in evaluation protocols – extended results

In the main paper in Tables 2, 3, and 4, we showed how changing the evaluation protocols impact the PSNR. In this section, we further show the impact on SSIM and LPIPS.

**Mip-NeRF 360 dataset.** For the Mip-NeRF 360 dataset, we suggested changing the way images are downscaled which yields different results (*cf.* Section 5.1 'Importance of unified evaluation protocol' in the main paper). We showed that by using the original large-scale images and downscaling them manually, the PSNR can improve such that if only a subset of methods use it, the ranking would change. In Table 6, we show the SSIM [6] and LPIPS [79] (VGG) averaged over all scenes of the Mip-NeRF 360 dataset under protocols $P_1$ (official NerfBaselines one), and $P_2$ (when manually downscaling the large images). The results are similar as for the PSNR, but we can see that LPIPS is more robust to the change. The relative order is the same within each evaluation protocol and the results do not change as much as in PSNR or SSIM cases. This can be attributed to LPIPS processing higher-level features instead of working on a pixel level.

| | \multicolumn{6}{SSIM} | | | | | | \multicolumn{6}{LPIPS (VGG)} | | | | | |
|---|---|---|---|---|---|---|---|---|---|---|---|---|
| | $P_1$ SSIM | $P_1$ rank | $P_2$ SSIM | $P_2$ rank | $P_2$ one-in | $P_1$ one-in | $P_1$ LPIPS | $P_1$ rank | $P_2$ LPIPS | $P_2$ rank | $P_2$ one-in | $P_1$ one-in |
|---|---|---|---|---|---|---|---|---|---|---|---|---|
| Zip-NeRF [4] | 0.829 | 1 | 0.839 | 1 | 1 | **2** | 0.218 | 1 | 0.206 | 1 | 1 | 1 |
| Scaffold-GS [37] | 0.813 | 6 | 0.824 | **5** | **3** | 6 | 0.262 | 6 | 0.248 | 6 | **3** | 6 |
| Mip-Splatting [75] | 0.815 | 3 | 0.826 | 3 | **2** | **6** | 0.258 | 5 | 0.245 | 5 | **3** | **6** |
| Gaussian Splatting [23] | 0.814 | 5 | 0.824 | 5 | **3** | **6** | 0.257 | 4 | 0.244 | 4 | **3** | **6** |
| Gaussian Opacity Fields [76] | 0.826 | 2 | 0.836 | 2 | **1** | 2 | 0.234 | 2 | 0.222 | 2 | 2 | 2 |
| gsplat [71] | 0.815 | 3 | 0.825 | **4** | 3 | **6** | 0.256 | 3 | 0.243 | 3 | 3 | **6** |
| 2D Gaussian Splatting [22] | 0.796 | 7 | 0.807 | 7 | 7 | 7 | 0.297 | 7 | 0.284 | 7 | 7 | 7 |
| NerfStudio [60] | 0.731 | 8 | 0.744 | 8 | 8 | 8 | 0.343 | 8 | 0.331 | 8 | 8 | 8 |
| Instant-NGP [46] | 0.684 | 9 | 0.703 | 9 | 9 | 9 | 0.398 | 9 | 0.380 | 9 | 9 | 9 |

Table 6: **Mip-NeRF 360 evaluation protocol differences.** We report SSIM and LPIPS under protocols $P_1$ (official), and $P_2$ (manual downscaling). Bold numbers denote changes in ranking.

**Blender dataset.** The Blender dataset [44] contains RGBA images with a transparent background. In the original NeRF paper, the authors used white color as the background ($P_1$). However, in Instant-NGP, a black background color was used ($P_2$). In Table 3 in the main paper, we showed the PSNR averaged over all Blender scenes. Here, in Table 7, we also show the SSIM [6] and LPIPS [79] (VGG). For these metrics, the results are very saturated and robust to the change of background. There are only tiny differences under the different protocols and the ranking is mostly kept the same.

| | \multicolumn{6}{SSIM} | | | | | | \multicolumn{6}{LPIPS (VGG)} | | | | | |
|---|---|---|---|---|---|---|---|---|---|---|---|---|
| | $P_1$ SSIM | $P_1$ rank | $P_2$ SSIM | $P_2$ rank | $P_2$ one-in | $P_1$ one-in | $P_1$ LPIPS | $P_1$ rank | $P_2$ LPIPS | $P_2$ rank | $P_2$ one-in | $P_1$ one-in |
|---|---|---|---|---|---|---|---|---|---|---|---|---|
| Gaussian Opacity Fields [76] | 0.969 | 1 | 0.970 | 1 | 1 | 1 | 0.038 | 2 | 0.037 | **1** | **1** | **1** |
| Mip-Splatting [75] | 0.969 | 1 | 0.969 | **2** | 1 | **2** | 0.039 | 3 | 0.039 | 3 | 3 | 3 |
| Gaussian Splatting [23] | 0.969 | 1 | 0.969 | **2** | 1 | **2** | 0.037 | 1 | 0.038 | **2** | 1 | 1 |
| Scaffold-GS [37] | 0.966 | 4 | 0.966 | 4 | 4 | 4 | 0.048 | 4 | 0.042 | 4 | 4 | 4 |
| Instant-NGP [46] | 0.959 | 5 | 0.959 | 5 | 5 | 5 | 0.055 | 5 | 0.054 | 5 | 5 | 5 |

Table 7: **Blender evaluation protocols comparison.** We report SSIM and LPIPS (VGG) under protocols $P_1$ (official), and $P_2$ (black background). Bold numbers denote changes in ranking.

**Photo Tourism dataset.** In the NeRF-W [40] paper, the evaluation protocol uses the left half of each test image to optimize the image's appearance embedding and to compute the metrics on the (previously unseen) right part. In Section 5.1 in the main paper (*cf.* Table 4), we compare this to a protocol where the appearance embedding of the test image is optimized on the full image [77, 66]. In Table 8, we show the SSIM and LPIPS (AlexNet) averaged over scenes Trevi Fountain, Brandenburg Gate, and Sacre Coeur of the Photo Tourism dataset [58]. Protocol $P_1$ is the official NerfBaselines one (NeRF-W [40]), and $P_2$ uses full test images when optimizing appearance embeddings. We can see, that both SSIM and LPIPS are more robust to the change of the protocol, where only the first three places are permuted for SSIM, and in the LPIPS case, everything is kept the same.

| | SSIM | | | | | | LPIPS (AlexNet) | | | | | |
|---|---|---|---|---|---|---|---|---|---|---|---|---|
| | $P_1$ SSIM | $P_1$ rank | $P_2$ SSIM | $P_2$ rank | $P_2$ one-in | $P_1$ one-in | $P_1$ LPIPS | $P_1$ rank | $P_2$ LPIPS | $P_2$ rank | $P_2$ one-in | $P_1$ one-in |
| WildGaussians [31] | 0.851 | 3 | 0.855 | 3 | **2** | 3 | 0.179 | 3 | 0.177 | 3 | 3 | 3 |
| gsplat [71] | 0.857 | 1 | 0.872 | 1 | 1 | **2** | 0.162 | 1 | 0.156 | 1 | 1 | 1 |
| Scaffold-GS [37] | 0.854 | 2 | 0.862 | 2 | **1** | **3** | 0.170 | 2 | 0.164 | 2 | 2 | 2 |
| NeRF-W *re.* [40] | 0.790 | 5 | 0.806 | 5 | 5 | 5 | 0.268 | 5 | 0.251 | 5 | 5 | 5 |
| GS-W [77] | 0.817 | 4 | 0.830 | 4 | 4 | 4 | 0.213 | 4 | 0.200 | 4 | 4 | 4 |
| K-Planes [17] | 0.761 | 6 | 0.778 | 6 | 6 | 6 | 0.313 | 6 | 0.292 | 6 | 6 | 6 |

Table 8: **Photo Tourism evaluation protocol differences.** We report SSIM, LPIPS (AlexNet) under protocols $P_1$ (NeRF-W), and $P_2$ (full test images). Bold numbers denote changes in ranking.

## A.4  Extended results on integrated methods

We extend the results from Fig. 3 and 6 to give exact numbers on all datasets: Mip-NeRF 360 [3], Blender [44], and [28].

| Method | NerfBaselines | | | Paper | | | Runtime | |
|---|---|---|---|---|---|---|---|---|
| | PSNR↑ | SSIM↑ | LPIPS↓ | PSNR↑ | SSIM↑ | LPIPS↓ | Time | GPU mem. |
| Zip-NeRF [4] | 28.553 | 0.829 | 0.218 | 28.54 | 0.828 | 0.189 | 5h 30m 20s | 26.8 GB |
| Scaffold-GS [37] | 27.714 | 0.813 | 0.262 | – | – | – | 23m 28s | 8.7 GB |
| Mip-NeRF 360 [3] | 27.681 | 0.792 | 0.272 | 27.69 | 0.792 | 0.237 | 30h 14m 36s | 33.6 GB |
| 3DGS-MCMC [26] | 27.571 | 0.798 | 0.281 | – | – | – | 35m 8s | 21.6 GB |
| Mip-Splatting [75] | 27.492 | 0.815 | 0.258 | 27.79* | 0.827* | 0.203* | 25m 37s | 11.0 GB |
| Gaussian Splatting [23] | 27.434 | 0.814 | 0.257 | 27.20 | 0.815 | 0.214 | 23m 25s | 11.1 GB |
| Gaussian Opacity Fields [76] | 27.421 | 0.826 | 0.234 | – | – | – | 1h 3m 54s | 28.4 GB |
| gsplat [71] | 27.412 | 0.815 | 0.256 | – | – | – | 29m 19s | 8.3 GB |
| 2D Gaussian Splatting [22] | 26.815 | 0.796 | 0.297 | 27.04* | 0.805* | – | 31m 10s | 13.2 GB |
| NerfStudio [60] | 26.388 | 0.731 | 0.343 | – | – | – | 19m 30s | 5.9 GB |
| Instant NGP [46] | 25.507 | 0.684 | 0.398 | – | – | – | 3m 54s | 7.8 GB |
| COLMAP [57] | 16.670 | 0.445 | 0.590 | – | – | – | 2h 52m 55s | – |

Table 9: **Mip-NeRF 360 [3] results.** We show the PSNR, SSIM [6], and LPIPS [79] (VGG) of various implemented methods averaged over all Mip-NeRF 360 [3] scenes. We also report the numbers from the papers. * methods used a different image downscaling method (*cf.* Section 5.1 in the main paper). The first , second , and third values are highlighted.

| Method | NerfBaselines | | | Paper | | | Runtime | |
|---|---|---|---|---|---|---|---|---|
| | PSNR↑ | SSIM↑ | LPIPS↓ | PSNR↑ | SSIM↑ | LPIPS↓ | Time | GPU mem. |
| Zip-NeRF [4] | 33.670 | 0.973 | 0.036 | 33.09 | 0.971 | 0.031 | 5h 21m 57s | 26.2 GB |
| Gaussian Opacity Fields [76] | 33.451 | 0.969 | 0.038 | – | – | – | 18m 26s | 3.1 GB |
| Mip-Splatting [75] | 33.330 | 0.969 | 0.039 | – | – | – | 6m 49s | 2.7 GB |
| Gaussian Splatting [23] | 33.308 | 0.969 | 0.037 | 33.31 | | – | 6m 6s | 3.1 GB |
| TensoRF [8] | 33.172 | 0.963 | 0.051 | 33.14 | 0.963 | 0.047 | 10m 47s | 16.4 GB |
| Scaffold-GS [37] | 33.080 | 0.966 | 0.048 | 33.68 | – | – | 7m 4s | 3.7 GB |
| 3DGS-MCMC [26] | 33.068 | 0.969 | 0.040 | 33.80† | 0.970† | – | 6m 13s | 3.9 GB |
| K-Planes [17] | 32.265 | 0.961 | 0.062 | – | – | – | 23m 58s | 4.6 GB |
| Instant NGP [46] | 32.198 | 0.959 | 0.055 | 31.18* | – | – | 2m 23s | 2.6 GB |
| Tetra-NeRF [30] | 31.951 | 0.957 | 0.056 | 31.52 | 0.982 | – | 6h 53m 20s | 29.6 GB |
| gsplat [71] | 31.471 | 0.966 | 0.054 | – | – | – | 14m 45s | 2.8 GB |
| Mip-NeRF 360 [3] | 30.345 | 0.951 | 0.060 | – | – | – | 3h 29m 39s | 114.8 GB |
| NerfStudio [60] | 29.191 | 0.941 | 0.095 | – | – | – | 9m 38s | 3.6 GB |
| COLMAP [57] | 12.123 | 0.766 | 0.214 | – | – | – | 1h 20m 34s | – |

Table 10: **Blender [44] results.** We show the PSNR, SSIM [6], and LPIPS [79] (VGG) of various implemented methods averaged over all Blender [44] scenes. We also report the numbers from the papers. † black background was used for blending instead of white; * exact hyperparameters for the dataset were not released at the time of writing. The first , second , and third values are highlighted.

**Mip-NeRF 360 [3].** First, we show the full results on the Mip-NeRF 360 [3] dataset. We show the PSNR, SSIM [6], LPIPS [79] (VGG), and the total NVIDIA A100 GPU memory used, as well as the total training time. The results, averaged over all scenes, are given in Table 9.

| SSIM↑ | barn | caterpillar | truck | lighthouse | playground | train | auditorium | ballroom | courtroom | museum | palace | temple |
|---|---|---|---|---|---|---|---|---|---|---|---|---|
| | | Training Data | | | Intermediate | | | | Advanced | | | |
| Instant NGP [46] | 0.772 | 0.633 | 0.770 | 0.765 | 0.696 | 0.657 | 0.761 | 0.652 | 0.640 | 0.471 | 0.668 | 0.689 |
| NerfStudio [60] | 0.794 | 0.666 | 0.797 | 0.768 | 0.755 | 0.693 | 0.771 | 0.705 | 0.673 | 0.648 | 0.640 | 0.678 |
| Zip-NeRF [4] | 0.884 | 0.802 | 0.864 | 0.849 | 0.880 | 0.814 | 0.877 | 0.835 | 0.790 | 0.746 | 0.718 | 0.805 |
| Gaussian Splatting [23] | 0.852 | 0.791 | 0.853 | 0.843 | 0.848 | 0.791 | 0.871 | 0.824 | 0.790 | 0.764 | 0.736 | 0.806 |
| Mip-Splatting [75] | 0.855 | 0.790 | 0.857 | 0.844 | 0.861 | 0.795 | 0.872 | 0.826 | 0.791 | 0.768 | 0.731 | 0.805 |
| Gaussian Opacity Fields [76] | 0.866 | 0.791 | 0.860 | 0.833 | 0.869 | 0.796 | 0.871 | 0.818 | 0.781 | 0.761 | 0.683 | 0.794 |
| | | | | | | | | | | | | |
| **LPIPS↑** | | | | | | | | | | | | |
| Instant NGP [46] | 0.271 | 0.360 | 0.216 | 0.281 | 0.343 | 0.334 | 0.429 | 0.352 | 0.448 | 0.606 | 0.440 | 0.424 |
| NerfStudio [60] | 0.215 | 0.302 | 0.167 | 0.245 | 0.249 | 0.261 | 0.330 | 0.261 | 0.336 | 0.311 | 0.452 | 0.392 |
| Zip-NeRF [4] | 0.083 | 0.152 | 0.081 | 0.131 | 0.095 | 0.119 | 0.153 | 0.113 | 0.153 | 0.159 | 0.317 | 0.183 |
| Gaussian Splatting [23] | 0.160 | 0.190 | 0.108 | 0.156 | 0.170 | 0.171 | 0.193 | 0.101 | 0.165 | 0.160 | 0.350 | 0.222 |
| Mip-Splatting [75] | 0.161 | 0.197 | 0.109 | 0.159 | 0.155 | 0.172 | 0.196 | 0.098 | 0.165 | 0.158 | 0.354 | 0.226 |
| Gaussian Opacity Fields [76] | 0.140 | 0.187 | 0.099 | 0.181 | 0.142 | 0.164 | 0.194 | 0.107 | 0.168 | 0.152 | 0.443 | 0.234 |

Table 11: **Tanks & Temples [28] results.** We show the SSIM [6], and LPIPS [79] (AlexNet) of various implemented methods. The first , second , and third values are highlighted.

**Extended results on Blender [44]** Similarly, we show the full results on the Blender [44] dataset. We show the PSNR, SSIM [6], LPIPS [79] (VGG), and the total NVIDIA A100 GPU memory used, as well as the total training time. The results, averaged over all scenes, are given in Table 10.

**Extended results on Tanks and Temples [28]** In the main paper (*cf.* Section 5.3), we show the PSNR for the Tanks and Temples dataset [28] for various methods. Here we also show the SSIM [6], and LPIPS [79] (AlexNet). The results are given in Table 11.

## A.5  Web platform

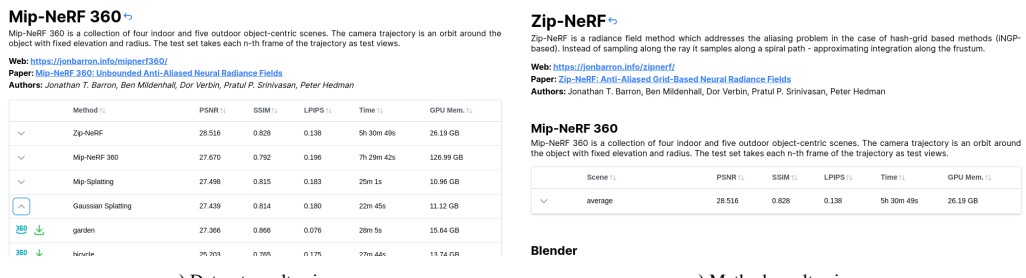

**a)** Dataset results view          **a)** Method results view

Figure 7: **Web platform.** Shows the ranking of the current set of integrated methods. It enables downloading of the checkpoints and predictions, and for some methods, it provides an online viewer.

To keep track of the current state of the art (SoTA), we release a web platform. The web platform shows results on all individual scenes for all methods, enables comparing methods, and allows users to download checkpoints and predictions for the datasets. Example screenshots can be seen in Figure 7.

## A.6  Use instructions

Before installing NerfBaselines, Python 3.7+ must be installed on the host system. We recommend using either `conda` or `venv` to separate NerfBaselines from system packages. After Python is ready, install the `nerfbaselines` pip package on your host system by running: `pip install nerfbaselines`. Now, `nerfbaselines` CLI (command line interface) can be used to interact with NerfBaselines. However, at least one supported backend must be installed before any method can be used. At the moment there are the following backends implemented:

- **docker**: Offers good isolation, requires `docker` (with NVIDIA container toolkit) to be installed and the user to have access to it (being in the docker user group). In order to install

it, please follow the instructions at [https://github.com/NVIDIA/nvidia-container-toolkit](https://github.com/NVIDIA/nvidia-container-toolkit)

- **apptainer**: Similar level of isolation as `docker`, but does not require the user to have privileged access. To install the backend, please follow instructions at [https://apptainer.org/docs/admin/main/installation.html](https://apptainer.org/docs/admin/main/installation.html).
- **conda** (default): Does not require `docker/apptainer` to be installed, but does not offer the same level of isolation and some methods require additional dependencies to be installed. Also, some methods are not implemented for this backend because they rely on dependencies not found on `conda`. To install `conda`, we recommend following instructions at [https://github.com/conda-forge/miniforge](https://github.com/conda-forge/miniforge) to install the `miniforge` distribution of `conda`.
- **python**: Will run everything directly in the current environment. Everything needs to be installed in the environment for this backend to work.

Additionally, all backends require NVIDIA GPU drivers to be installed to access the GPUs. For NerfBaselines commands, the backend can be set either via the `--backend <backend>` argument or using the `NERFBASELINES_BACKEND` environment variable.

**Training.** To start the training, use the following command: `nerfbaselines train --method <method> --data external://<dataset>/<scene>`, where `<method>` can be e.g., `nerfacto`, `zipnerf`, `instant-ngp`, ... (for the full list, run `nerfbaselines train --help`). The `<dataset>` can be one of the following: `mipnerf360`, `blender`, `tanksandtemples`, *etc.*. Similarly, `<scene>` is the scene name in lowercase. The training script will automatically download the dataset and start the training. The training will also run the evaluation and output the metrics computed on the test set.

**Other commands.** The resulting checkpoint can be used in the viewer (`nerfbaselines viewer --checkpoint <checkpoint> --data external://<dataset>/<scene>`), to rerun the rendering (`nerfbaselines render --checkpoint <checkpoint> --data external://<dataset>/<scene>`), or to render a camera trajectory (`nerfbaselines render-trajectory --checkpoint <checkpoint> --trajectory <trajectory> --output <output>.mp4`). The full list of available commands can be seen by running `nerfbaselines --help`.

