# OpenReview forum: "NerfBaselines: Consistent and Reproducible Evaluation of Novel View Synthesis Methods"
_NeurIPS.cc/2025/Datasets_and_Benchmarks_Track — NeurIPS 2025 Datasets and Benchmarks Track poster_

### Official Review · Reviewer_My4t · 2025-06-25

**Rating:** 5
**Confidence:** 4

**Summary:**

The paper introduces a "NerfBaselines" which is a unified evaluation framework designed for benchmarking the novel view synthesis approaches, including NeRF- and Gaussian-Splatting-based methods. The authors aim to ensure a consistent, fair, and reproducible evaluation. This includes investigating the current evaluation protocols, providing a unified wrapper for various models, and a viewer for visual inspection of the results.

**Additional Feedback:**

Referring to adoption by the community, are the authors able to roughly quantify the workload required to get an arbitrary method compatible with the framework (I assume this includes unifying data convention between NerfBaselines and the existing code)?

A suggestion regarding adoption by the community would be to consider partnering or running a challenge associated with a workshop, which could encourage the use of the framework.

Regarding other scenarios of reconstruction, are there any plans to include sparse/mono methods/evaluations, or dynamic ones?

The authors mention the use of appearance embeddings. Would it be easy to introduce other types of modalities into the representation (e.g. CLIP embeddings, such as in LangSplat)?

**Dataset Code Accessibility:**

Yes

**Dataset Code Comments:**

The code is available and includes all the methods mentioned in the paper. Similarly, the website is live, and all the results are available to either download or preview in the viewer.

**Ethical Considerations:**

No, there are no or only very minor ethics concerns

**Final Justification:**

I am convinced by the authors' rebuttal; my concerns were addressed. I appreciate the work towards including more NVS tasks. I know of researchers already using the NerfBaselines, hence, it eases my worries about adoptability.

**Limitations Weaknesses:**

I believe the main weakness of the paper was mentioned by the authors themselves in the Limitations section. Namely, the success of the framework is directly dependent on its wider adoption by the community.  To this end, I would like to see some sort of tutorial on how to convert an existing method into a NerfBaselines-compatible format (as opposed to writing it in a compatible form from scratch).

While I strongly appreciate the study on the evaluation protocol, I would like to see more discussion on what the best practice is while evaluating the rendered images. NerfBaselines aims to establish a uniform evaluation; hence, some design choices could be discussed. For example, what is the appropriate way to quantise float renderings (cast to int vs round and cast - as both are seen in some repositories), or when the dataset is released in one resolution but typically methods downscale the data, what is the appropriate approach.

Another point I would like to raise is the shift in popularity between typical multi-view scenarios in favour of constrained scenarios (e.g. sparse or mono). This is not currently included in the framework. Similarly, current research interest is heavily oriented towards dynamic modelling. I do not know how easy or hard it would be to include such methods/datasets into the framework (as they are typically a bit more intricate).

**Strengths Contributions:**

I believe the paper is well written and offers an interesting contribution to the research community.

The initial part of the work focuses on comparing existing codebases. I believe the comparison is fair and sufficiently points out differences and shortcomings in the compared codebases. I find comparison to NerfStudio particularly useful in emphasising differences to NerfBaselines as it can be considered the closest competitor. Notably, NerfBaselines isolates the model implementation, whereas NerfStudio reimplements the included methods. This is an important factor, as researchers would not be inclined to use an underperforming implementation.

Similarly, regarding the NerfBaselines codebase, it includes a reasonable number of existing methods.  The authors commit to supporting the extension of the platform, which, in my belief, is crucial for such a project.

A second major part of this paper is the analysis of the existing evaluations. I believe it is an important and often overlooked topic in the community. In the age of squeezing out every bit of performance metric, a scrutiny of the evaluation protocol should not be a neglected topic. To this end, the authors compared the evaluation of several datasets across several methods. The main observations include noticing that changing a seemingly minor detail in the evaluation (such as when the image is downscaled) can make the method appear to outperform its competitors. This goes to a strength of NerfBaselines, as it prevents misleading evaluations due to the protocol being tied to the dataset within the framework.

Further, what is not as strongly emphasised in the paper itself but I find to be particularly useful, is the existence of the leaderboard website. It is already maintained and accessible, containing the results from this work. I find a collection of results across the methods gathered in one place useful, e.g. for streamlining the review process (as a trustworthy source to compare the numbers reported in the papers).

As a minor advantage, it is worth mentioning that the framework provides several ways of running the methods (either conda or containerisation), which conform to standards and common practices.

---

> ### Author Rebuttal · Authors · 2025-07-31
>
> We thank the reviewer for the constructive feedback and will adjust the paper based on their comments and the rebuttal.
>
> **\[W1\] Success of framework depends on adoption by community**
> Indeed, as is the case with any benchmarking dataset / framework, the success of the framework will in the end depend on the adoption by the community. Naturally, this is hard to predict. We believe the technical side and ease-of-use of our project make it a strong candidate for a wide adoption by the community. We believe that the framework is useful for the community and thus deserves the chance of being discovered by the community (and a potential NeurIPS datasets & benchmark publication would significantly help with this)
>
> **\[W1\] Tutorial on integrating methods into a NerfBaselines**
> The existing tutorial on the project documentation webpage (under “Adding new methods”) describes integrating a fictional method. We can extend/modify the tutorial to show how to integrate an actual existing method instead of a fictional one.
>
> **\[W2\] Evaluation protocols design choices**
> We would like to argue that many of the evaluation protocol choices can be similarly valid and it cannot be said that one is better than others from the perspective of comparing different methods. For example, the background color used in the Blender dataset can be white or black or some other color and it can really be arbitrary. This holds for other choices as well, e.g., computing metrics on quantized images in uint8 range or on fp32 data is both viable. Notice that relative order is mostly preserved under the “P2 rank” in Tables 2, 3 under “sensible” choices for the evaluation protocol (such as the image downscaling method or the dataset background color). The key takeaway we wanted to highlight in the paper is to ensure that the same evaluation protocol is used for all methods, which is not the case with public implementations and results reported in the literature.
>
> Even though for some of the design choices, many options can be justified to be equally good for comparing different methods, practicality can be a big deciding factor for the design choices. One important aspect is to make the protocol as consistent with most published numbers as possible. This avoids creating confusion in the community with different sets of numbers and allows us to verify and validate the numbers reported in the literature. We now give detailed motivation for the design choices taken in our evaluation protocols:
>
> 6. **Images are evaluated in the uint8 range.** The reason is that for reproducibility, every published work should store and publish predictions on the test set. This enables the results to be verified and also enables future computation of new metrics and qualitative comparison, etc. These predictions are stored in uint8 which is sufficient for most datasets (consider uint8 images were used for training) and fp32 images would require significantly more memory. Furthermore, almost all publications released either none or uint8 image predictions and evaluation protocols should be compatible with this design choice to be able to use this data and compare with these methods. Therefore, during the evaluation, images have to be rounded to the uint8 range because the predictions will later be stored as uint8 and the predictions must be reproducible from the released images.
> 7. **Using AlexNex LPIPS.** We see a trend in recent literature of people computing LPIPS with the AlexNet network. The benefit is that AlexNet is more lightweight and it makes evaluation faster.
> 8. **Using pre-downscaled images for Mip-NeRF 360\.** We argue for the use of pre-downscaled released images, because the default downscaling algorithms differ for different platforms and libraries. Loading larger images and downscaling them before evaluation could yield different results depending on platform \- which is exactly what the unified evaluation protocol should avoid.
> 9. **Using white background for the Blender dataset.** We want to have consistency with as large a number of papers from the literature as possible, hence we follow the same design choice.
> 10. **SSIM defaults.** Again, many choices are sensible and we just need to make sure the exact same code and parameters are used. This is not the case with some existing libraries (e.g., torchmetrics, dmpix, and skimage), where the defaults can differ. Our choice (default for skimage) is predominantly used in the literature, hence we follow the same practise.
>
> We thank the reviewer for raising this concern and will extend the paper with the reasoning behind the design choices for the evaluation protocols.
>
> **\[Q1\] Quantify workload required to integrate new method**
> Usually it takes between 2-5 hours for new methods with a good codebase (\<2years old), and 10-24 hours for old methods or poorly-written code to be integrated into NerfBaselines. Older methods require a longer time due to the need for getting dependencies to install reliably and poorly-written code takes more time to patch in order to be accessible from the API. Integration includes writing a compatibility layer to translate between the data loading code and NerfBaselines dataset, writing a robust install script to install dependencies, and testing the integrated method. For older methods, it can take time to figure out how to install them as some dependencies / packages might no longer exist and the original installation instructions might no longer hold. A tutorial on how to integrate new methods can be found in the official documentation under “Adding new methods”.
>
> As an example, for the rebuttal, we integrated 2 more methods based on 3DGS, which took 2 and 4 hours respectively to
>
> 1) write an install script \- this includes figuring out which packages were not mentioned in READM/requirements.txt and find correct version to install compatible with the rest of the packages,
> 2) integrate the NerfBaselines data format into the method,
> 3) fix the methods to work with more general cameras,
> 4) extend the method to support alpha masks,
> 5) write unit tests for the method \- this mainly involves writing a code to mock dependencies, e.g. mocking a renderer in the case of 3DGS,
> 6) write metadata like paper title, licenses, etc.,
> 7) test the method with automated testing tools, and
> 8) train the method on 18 scenes (running in parallel).
>
> **\[Q2\] A suggestion regarding adoption by the community would be to consider partnering or running a challenge associated with a workshop, which could encourage the use of the framework.**
> Thank you very much for the suggestion. Running a challenge is a great idea which would indeed boost the project’s visibility. It is something we are also considering.
>
> **\[W3, Q3\] Plans to include sparse/mono NVS or dynamic scenes?**
> **Adding sparse-view NVS task**
> Thank you for the suggestion. We agree with the reviewer that including more tasks will make the benchmark more versatile. While developing NerfBaselines, we focused our attention on getting a large number of NVS methods integrated first before expanding into other subdomains. However, now we have \~28 NVS methods integrated and decided to expand into more datasets/tasks. For the rebuttal, we have added a new task \- sparse novel view synthesis (as requested) and integrated one sparse-view dataset (sparse-view MipNeRF 360) and two sparse-view NVS methods (DropGaussian, SparseGS) to demonstrate the scalability. The results can be seen on the webpage (links omitted as per NeurIPS policy) and in the table below. We aim to continue extending NerfBaselines with more tasks/methods.
>
> | Method | PSNR | SSIM | LPIPS (VGG) | Time | GPU mem. |
> | :---- | ----: | ----: | ----: | ----: | ----: |
> | DropGaussian | **21.452** | **0.616** | 0.458 | **5m 21s** | **4.8 GB** |
> | Scaffold-GS | *20.282* | *0.585* | *0.410* | 20m 41s | *5.6 GB* |
> | 3DGS-MCMC | 20.068 | 0.578 | **0.407** | 35m 60s | 23.0 GB |
> | Gaussian Splatting | 19.758 | 0.568 | 0.412 | *19m 46s* | 7.2 GB |
> | Mip-Splatting | 19.750 | 0.572 | 0.412 | 20m 19s | 6.8 GB |
> | SparseGS | 19.646 | 0.573 | 0.438 | 42m 5s | 11.6 GB |
> | PGSR | 17.199 | 0.491 | 0.468 | 26m 27s | 6.9 GB |
> | Zip-NeRF | 16.466 | 0.428 | 0.535 | 5h 27m 11s | 29.8 GB |
>
> **Adding dynamic scenes**
> Dynamic scenes are supported by the current API. The data type representing cameras allows for additional metadata which can be the time dimension in this case. The metadata is supported throughout the API including training, rendering, and the viewer. However, there are currently no dynamic NVS methods or datasets integrated. We plan to integrate dynamic NVS datasets and methods in the near future.
>
> **\[Q4\] Adding other modalities such as CLIP embeddings**
> Yes, the appearance embeddings can be arbitrary and other types of modalities can easily be supported by the current API. However, mapping from texts to embeddings is not supported at the moment and it will require extending the API and the viewer (with a textbox). Still, if there is sufficient interest in text-based appearance description, we will consider implementing it. Currently, the feedback from the community helps us prioritize which tasks/methods to implement/integrate next.

---

> > ### Comment · Reviewer_My4t · 2025-08-03
> > **Rebuttal response**
> >
> > I thank the authors for their detailed rebuttal. I appreciate the answers, which lead me to believe that NerfBaselines can and will be extended by the authors, including the addition of new tasks. Regarding the protocol design choices, I completely agree that some choices are equally viable and not quantifiably better. My suggestion is to include those choices and motivation in the paper to set a clearer standard for the community.

---

> > > ### Author Response · Authors · 2025-08-05
> > >
> > > Thank you for your suggestion! As we promised in our rebuttal, we will extend the paper with the motivation for the design choices (W2) namely: using uint8 range for image evaluation, using AlexNet LPIPS for new datasets, using pre-downscaled images, white background for Blender dataset, and details on SSIM parameters. Is there something we can add/clarify so that you would consider raising the score to Accept?

---

> > > > ### Comment · Reviewer_My4t · 2025-08-06
> > > >
> > > > I am currently inclined to increase the score based on the rebuttal. I would like to first take into account the coming reviewers' discussion.

---

### Official Review · Reviewer_Fkoi · 2025-07-03

**Rating:** 5
**Confidence:** 3

**Summary:**

The paper highlights an important problem in research in general: different research works can be using slightly different evaluation protocols, potentially impacting the overall comparability of the results. In particular, the paper focuses on the domain of novel view synthesis, and proposes a framework to evaluate NVS methods with a repeatable protocol. The paper describes the framework, in which methods do not have to be fully reimplemented but can be adapted from the original code, and validates it with experiments on existing methods. Results show that existing findings can be mostly trusted, but slight changes in the evaluation protocol can invert some of them, showcasing the importance of a unified benchmarking framework.

**Dataset Code Accessibility:**

Yes

**Ethical Considerations:**

No, there are no or only very minor ethics concerns

**Final Justification:**

I have carefully read the other reviews and the rebuttals. I believe this paper presents a very useful and well thought-out framework, which already has a good user base, and deserves acceptance. The authors provided satisfying answers both to my points and to the points of other reviewers, in my opinion.

**Limitations Weaknesses:**

I do not currently have major concerns with this work. The only weakness I identified lies in the fact that frameworks for developing and testing NVS methods already exist, so creating a new one can fragment pipelines even more instead of becoming a globally accepted standard. Existing frameworks have their own flaws (for example Nerf-Studio requires methods to be implemented in a specific way), which this work aims to fix or alleviate, but it is always difficult to become a standard.
The proposed framework, however, appears solidly thought out to me, so I do not have concerns on the technical side of it.

**Strengths Contributions:**

S1) The paper is well written and easy to follow
S2) The motivations behind the paper are well justified and, while existing findings can be trusted for the most part, it remains important to have common evaluation protocol
S3) The experiments show that the code adaptations from original repositories achieve comparable scores to the originals, which is not so easy to achieve when a full reimplementation is required

---

> ### Author Rebuttal · Authors · 2025-07-31
>
> We thank the reviewer for the constructive feedback and will adjust the paper based on their comments and the rebuttal.
>
> ### **\[W1\] Frameworks for developing NVS methods already exist**
>
> While there exist frameworks for developing NVS methods (e.g. Nerfstudio), the goal of our method is different. Nerfstudio provides a toolbox for developing new methods \- including existing dataloaders, viewer, and various building blocks. On the other hand, the focus of our method is solely on benchmarking other approaches and to enable evaluating/interacting with other methods developed in different frameworks in an unified way. We do not provide building blocks nor encourage users to build their methods on top of NerfBaselines. However, we provide a toolbox for benchmarking methods in a safe, reproducible way \- we ensure exact and safe evaluation protocols are used, reproducible checkpoints, we provide an interactive viewer, and a tool to automatically build a web benchmark. Our main design principle is to make integration of new methods as easy as possible. For that reason, for example, we do not even enforce the method to run under the same interpreter because some old methods (e.g., NeRF) require older Python versions with TensorFlow 1.x, which would not be compatible with modern Python environments, and we isolate the method into a container connected with inter-process communication. This is in contrast to existing frameworks which require implemented methods to share environment, API, and inherit framework’s base classes, etc. Furthermore, we maintain an online benchmark where current methods are compared and where users can contribute their methods. This is very useful for research and we have very positive feedback from the community (e.g. on X, but we cannot include links by NeurIPS policy). Having a public benchmark is something which is missing in NVS frameworks.

---

> > ### Comment · Reviewer_Fkoi · 2025-08-05
> >
> > I thank the authors for the detailed response. I have carefully read the other reviews and the rebuttals, and I intend to maintain my rating unless something major comes up during discussion. I believe the framework to be very useful and the paper to deserve acceptance.

---

### Official Review · Reviewer_kouB · 2025-07-03

**Rating:** 4
**Confidence:** 4

**Summary:**

This work proposes a unified evaluation protocol to evaluate novel-view synthesis methods. The unified evaluation protocol includes background color for segmented images, downscaling methods, order between image type (uint8 vs float), etc. The authors found that different evaluation protocol produces big difference in rankings for some methods, which supports the claim of the necessity of unified evaluation protocol. However, it was not clearly presented why it is more reasonable to follow the protocol suggested by NeRFBaselines when conducting evaluation.

**Additional Feedback:**

In general, I agree with the necessity of unified evaluation protocol for novel-view synthesis method. I'm willing to increase my score if the authors could provide more information to explain the reason to follow the criterion that NeRFBaselines chose.

**Dataset Code Accessibility:**

Yes

**Dataset Code Comments:**

The official document (https://nerfbaselines.github.io/) clearly describes how to evaluate new methods with NeRFBaselines benchmark. It should be straightforward to implement new method and evaluate within unified environment.

**Ethical Comments:**

I can't see any ethical concerns in this submission.

**Ethical Considerations:**

No, there are no or only very minor ethics concerns

**Final Justification:**

Please see my final comments to the rebuttal.

**Limitations Weaknesses:**

* My big concern is lack of justification of the proposed evaluation protocol. If small changes in the evaluation protocol choices makes big difference in the result, we need to unify the evaluation protocol "in a right way". However, there is no clear explanation for the reason why we need to follow the specific setups of NeRFBaselines selected.
* Containing other type of image sources would make this benchmark stronger. For example, authors could include 360 panorama images dataset and methods. (e.g., https://github.com/changwoonchoi/EgoNeRF)

**Strengths Contributions:**

* The authors show lots of methods can be integrated into NeRFBaselines evaluation code and the quantitative results also match the value from the original implementation.
* The motivation or necessity of unified evaluation protocol is clear. Also, the experiments show that small changes in the choice of evaluation protocol can make big difference of the rankings of methods, and the results strongly supports the necessity of the unified evaluation protocol.
* The benchmark is well documented and easy to integrate new methods.

---

> ### Author Rebuttal · Authors · 2025-07-31
>
> We thank the reviewer for the constructive feedback and will adjust the paper based on their comments and the rebuttal.
>
> ### **\[W1\] Reasoning behind design choices for evaluation protocols**
>
> We would like to argue that many of the evaluation protocol choices can be similarly valid and it cannot be said that one is better than others from the perspective of comparing different methods. For example, the background color used in the Blender dataset can be white or black or some other color and it can really be arbitrary. This holds for other choices as well, e.g., computing metrics on quantized images in uint8 range or on fp32 data is both viable. Notice that relative order is mostly preserved under the “P2 rank” in Tables 2, 3 under “sensible” choices for the evaluation protocol (such as the image downscaling method or the dataset background color). The key takeaway we wanted to highlight in the paper is to ensure that the same evaluation protocol is used for all methods, which is not the case with public implementations and results reported in the literature.
>
> Even though for some of the design choices, many options can be justified to be equally good for comparing different methods, practicality can be a big deciding factor for the design choices. One important aspect is to make the protocol as consistent with most published numbers as possible. This avoids creating confusion in the community with different sets of numbers and allows us to verify and validate the numbers reported in the literature. We now give detailed motivation for the design choices taken in our evaluation protocols:
>
> 1. **Images are evaluated in the uint8 range.** The reason is that for reproducibility, every published work should store and publish predictions on the test set. This enables the results to be verified and also enables future computation of new metrics and qualitative comparison, etc. These predictions are stored in uint8 which is sufficient for most datasets (consider uint8 images were used for training) and fp32 images would require significantly more memory. Furthermore, almost all publications released either none or uint8 image predictions and evaluation protocols should be compatible with this design choice to be able to use this data and compare with these methods. Therefore, during the evaluation, images have to be rounded to the uint8 range because the predictions will later be stored as uint8 and the predictions must be reproducible from the released images.
> 2. **Using AlexNex LPIPS.** We see a trend in recent literature of people computing LPIPS with the AlexNet network. The benefit is that AlexNet is more lightweight and it makes evaluation faster.
> 3. **Using pre-downscaled images for Mip-NeRF 360\.** We argue for the use of pre-downscaled released images, because the default downscaling algorithms differ for different platforms and libraries. Loading larger images and downscaling them before evaluation could yield different results depending on platform \- which is exactly what the unified evaluation protocol should avoid.
> 4. **Using white background for the Blender dataset.** We want to have consistency with as large a number of papers from the literature as possible, hence we follow the same design choice.
> 5. **SSIM defaults.** Again, many choices are sensible and we just need to make sure the exact same code and parameters are used. This is not the case with some existing libraries (e.g., torchmetrics, dmpix, and skimage), where the defaults can differ. Our choice (default for skimage) is predominantly used in the literature, hence we follow the same practise.
>
> We thank the reviewer for raising this concern and will extend the paper with the reasoning behind the design choices for the evaluation protocols.
>
> ### **\[W2\] Other datasets/tasks**
>
> We agree with the reviewer that including more data and more tasks will make the benchmark more versatile. While 360 panorama images are supported in our dataset API and can be loaded and used out of the box with NerfBaselines, there is currently no public dataset (and benchmark) integrated. While developing the method, we focused our attention on getting a large number of NVS methods integrated first before expanding into more specialized domains. However, now we have \~28 NVS methods integrated and decided to expand into more datasets/tasks. For the rebuttal, we have added a new task \- sparse novel view synthesis and integrated one sparse-view dataset and two sparse-view NVS methods to demonstrate the scalability (see reviewer My4t response W3,Q3). We aim to continue extending NerfBaselines with more tasks/methods.

---

### Official Review · Reviewer_oUuq · 2025-07-05

**Rating:** 4
**Confidence:** 5

**Summary:**

This dataset submission introduces a framework called **NerfBaselines**, aiming to address the issue of inconsistent evaluation protocols for NeRF (Neural Radiance Fields) and 3D Gaussian Splatting (3DGS) methods. Currently, most official implementations of these methods employ different dataset loaders, evaluation protocols, and metrics, making benchmarking difficult. NerfBaselines provides a unified interface for running and evaluating these methods on various datasets using consistent metrics. Instead of reimplementing the methods, the project leverages official implementations and wraps them to enable easy execution through a shared interface.
The paper highlights the importance of consistent and reproducible evaluations in the field of novel view synthesis and provides a valuable reference for researchers working on NeRF and 3DGS methods. By offering a unified evaluation framework and benchmarking results, the paper aims to promote the development and comparison of novel view synthesis techniques.

**Dataset Code Accessibility:**

Yes

**Ethical Considerations:**

No, there are no or only very minor ethics concerns

**Final Justification:**

Prefer to keep the final rate to be the same

**Limitations Weaknesses:**

This work while offering a unified framework for evaluating NeRF and 3DGS methods, also has some weaknesses compared to gsplat and Nerfstudio:
1. *Lower computational efficiency and speed*: Compared to Gsplat and Nerfstudio, NerfBaselines may have lower computational efficiency and speed. Gsplat is designed for rendering with Gaussian splatting, which can achieve faster rendering speeds and lower computational costs. Nerfstudio also has certain advantages in rendering speed and computational efficiency.
2. *Smaller community and fewer resources*: Compared to Gsplat and Nerfstudio, NerfBaselines may have a smaller community and fewer resources. Gsplat and Nerfstudio have garnered significant attention and usage within the research community, with extensive documentation, tutorials, and community support available. In contrast, the community size and resource availability for NerfBaselines may be relatively limited.
It should be noted that the weaknesses mentioned above are relative and may vary depending on specific usage scenarios and requirements.

**Strengths Contributions:**

1. The framework supports multiple backends, including Docker, Apptainer, Conda, and Python, offering users flexibility in choosing their preferred environment.
2. Additionally, it provides tools for automatic dataset downloading, training, rendering, and interactive visualization.
3. The paper includes benchmarking results for multiple methods on standard datasets such as Mip-NeRF 360, Blender, and Tanks and Temples. These results cover metrics like PSNR, SSIM, LPIPS, and computational resource usage (time and GPU memory). The authors aim to gradually expand the framework by implementing more methods.
4. The paper also provides detailed documentation and implementation status updates for each method, facilitating contributions from the research community.
5. An interactive viewer is provided

---

> ### Author Rebuttal · Authors · 2025-07-31
>
> We thank the reviewer for the constructive feedback and will adjust the paper based on their comments and the rebuttal.
>
> ### **\[W1\] Lower computational efficiency and speed**
>
> Both Gsplat and Nerfstudio are integrated into NerfBaselines, and therefore can be directly used from the CLI/API. Let us unpack how the rendering speed/computational costs relate and what are potential bottlenecks:
>
> * First, NerfBaselines integrates existing methods instead of reimplementing them. This means that when running Gsplat or Nerfacto from NerfBaselines, the user is really only invoking the original code and there is no difference in the code being run. In order for the methods to be used from NerfBaselines, a thin wrapper is written which delegates calls to the internal codebase and the wrapper adds basically no overhead (it is a single python call per training iteration).
> * Next, NerfBaselines installs each method inside a virtual environment (Conda/Docker/Apptainer). Inside this container, the set of dependencies is exactly the same as described in README/requirements.txt of each integrated method. Even the recommended Python version is matched. Therefore, there is no possible slowdown by using a possibly older PyTorch/JAX version.
> * There are two possible options when interacting with a method from the CLI. One is to launch a shell inside the container and run NerfBaselines from there, and the second one (described next) is to use inter-process communication. In the first case, there is no overhead both in terms of rendering/training speed or in terms of computational efficiency.
> * Finally, NerfBaselines also offers the calling process to be inside a separate environment. This is the default option as this enables users to install a single NerfBaselines package locally and to interface with different methods from the same environment/python session. In order to achieve this, NerfBaselines launches a worker process inside the containers and uses inter-process communication (IPC) protocols to delegate any calls from a “proxy” type into the real instance inside the container. To this end, NerfBaselines implements multiple communication protocols to achieve fast communication without significant overhead. The default protocol uses shared pipes to exchange pointers to shared memory. At inference time, after rendering an image, the rendered tensor needs to be copied from GPU to CPU. For TensorFlow or JAX methods, we copy the tensor from the GPU directly into CPU shared memory (inside the container process) and have a second copy in the host process. This can cause \<1ms overhead for very large tensors. For PyTorch-based methods (most methods), in order to remove the additional copy possibly caused by IPC, we exchange pointers to GPU memory between the container and the host process and only copy the data from GPU to CPU inside the host process. Therefore, there is no overhead compared to using CLI inside the container or using the original implementation directly.
>
> To summarize, NerfBaselines does not add significant overhead on top of integrated codebases. If inter-process communications are used for the added comfort, a tiny overhead \<1ms can occur on rendering for TensorFlow/JAX-based methods while no overhead should be observed for PyTorch.
>
> ### **\[W2\] Smaller community and fewer resources**
>
> **Gsplat and Nerfstudio are complementary rather than competitors**
> We agree there are projects with larger communities and more resources. However, we see NerfBaselines as complimentary to Gsplat and Nerfstudio rather than as a competitor:
> Both Gsplat and Nerfstudio are catering to communities that either want to apply these methods or want to build on top of the frameworks to develop novel methods. In contrast, NerfBaselines aims to enable fair comparisons between methods developed in different frameworks. I.e., NerfBaselines aims to enable a fair comparison between methods implemented in different frameworks by different communities. Furthermore, we maintain an online benchmark where current methods are compared and where users can contribute their methods. This is very useful for research and we have very positive feedback from the community (e.g. on X, but we cannot include links by NeurIPS policy). Having a public benchmark is something which is missing from Nerfstudio and other frameworks. Furthermore, both Gsplat and Nerfstudio are integrated in NerfBaselines \- they can be used from the CLI and API and their results are visible on our public benchmark. We aren’t aware of any other benchmarking NVS framework with public benchmark and a large community.
>
> **NerfBaselines has a small community**
> Compared to the Nerfstudio’s community which includes non-research people interested in running 3D reconstruction on casually captured data, NerfBaselines target audience is mainly people involved in NVS research. Therefore, we do not expect/aim to match Nerfstudio or other projects in the number of users. Yet, this does not mean that NerfBaselines is not useful. We argue that for the task of benchmarking NVS, NerfBaselines is very useful (as recognized by the reviewers) and has a growing community of researchers. While the current citation count is 10 (taken from Semantic Scholar) for the arxiv preprint (before being published), we see a fast growth in the total number of downloads (currently at 32,377 downloads) and already have 300 stars on github repository. We believe the project can reach a high popularity.

---

> > ### Comment · Reviewer_oUuq · 2025-08-05
> >
> > Given the potential of the community values, I'd like to keep the rates

---

> > > ### Author Response · Authors · 2025-08-05
> > >
> > > Thank you very much for acknowledging the value for the community! Is there something we can add/clarify so that you would consider raising the score to Accept?

---

### Decision · Program_Chairs · 2025-09-18

**Decision:**

Accept (poster)

**Comment:**

This submission received a mix of borderline and positive scores. Reviewers raised several common concerns, including:
* The difficulty of establishing the dataset as a community standard given the existence of Gsplat and NeRF-Factory.
* The need for stronger justification of the proposed evaluation protocols.
* The lack of extensions to broader tasks such as egocentric views and dynamic modeling.

Overall, the authors addressed these concerns thoughtfully. The Area Chair appreciates their efforts and hopes this review process will serve as an initial step toward unifying benchmarking practices in novel view synthesis.